# Population Dynamics of *Juniperus macropoda* Bossier Forest Ecosystem in Relation to Soil Physico-Chemical Characteristics in the Cold Desert of North-Western Himalaya

Dhirender Kumar [1], Daulat Ram Bhardwaj [1,†], Prashant Sharma [1,†], Bharti [2], Neeraj Sankhyan [3], Nadhir Al-Ansari [4] and Nguyen Thi Thuy Linh [5,*,†]

1   Department of Silviculture and Agroforestry, Dr YSP University of Horticulture and Forestry, Solan 173230, India
2   Division of Sample Survey, ICAR-Indian Agricultural Statistics Research Institute, New Delhi 110012, India
3   Department of Basic Science, Dr YSP University of Horticulture and Forestry, Solan 173230, India
4   Department of Civil, Environmental and Natural Resources Engineering, Lulea University of Technology, 97187 Lulea, Sweden
5   Institute of Applied Technology, Thu Dau Mot University, Thu Dau Mot City 75000, Binh Duong, Vietnam
*   Correspondence: nguyenthuylinh@tdmu.edu.vn
†   These authors contributed equally to this work.

**Abstract:** *Juniperus macropoda* is the only tree species of a cold desert ecosystem that is experiencing high anthropogenic pressure and has a poor regeneration status due to harsh environmental conditions. Due to the limited distribution of *Juniperus macropoda* in this region, the species have remained largely unexplored in terms of understanding the distribution pattern along the elevation and soil fertility gradients. Therefore, the current research was carried out along the elevational gradient, starting from the base line at 3000 m above sea level (m asl) asl with an elevational plot distance of 180 m. The study revealed that the average density of *J. macropoda* declined gradually from the first elevation range, i.e., 3000–3180 m asl onward, and extended up to the elevation range of 3900–4080 m asl. However, the average seedling and sapling densities were highest at mid-elevation and extended up to an elevation range of 4080–4260 m asl. The *J. macropoda* population formed a reverse J-shaped structure only up to 3540–3720 m asl. The maximum total biomass and carbon density were recorded in the lowest elevational range, and decreased subsequently. The primary soil nutrients under study decreased sharply along the elevational gradient. Seedling, sapling and tree distributions had a significantly positive relationship ($p < 0.05$) with available N, P, K, SOC, silt and clay contents and were negatively correlated ($p < 0.05$) with sand contents. The outcome of the study will form the basis for devising a plan for the management and conservation of *J. macropoda* forests.

**Keywords:** cold desert; soil nutrients; biomass; carbon density; elevational ranges; natural regeneration

## 1. Introduction

Mountains cover approximately 25% of the land surface area of the earth but host at least one-third of the terrestrial plant species diversity [1]. However, these high-mountain ecosystems are most fragile and vulnerable to the adverse impacts of climate change [2,3], resulting in a shift in the distribution of species [4–6]. The climatic factors, particularly temperature, humidity, and light, are widely regarded as the most important determinants of the distribution of plant species in the mountain ecosystem as they determine the local site conditions and abiotic niches [7–10].

However, besides the climatic factors, the site characteristics are one of the major aspects that influence the distribution of plant species on a local [11–14] and regional

scale [15–17] through the provision of nutrients and water in terms of changes in physico-chemical soil properties. On the local scale, the edaphic factors including soil pH, water-holding capacity, nutrient contents, slope and aspect govern seedling emergence, and establishment and subsequent growth and development influence the tree line position [18]. In contrast, on a broader scale, the soil nutrients [19] and water dynamics are regarded as the key regulators of ecosystem biomass and productivity [20–22], which determine the different forest vegetation cover [23,24]. Simultaneously, the soil nutrients play a profound role in the seedling establishment, tree growth and survival [25]. Therefore, over the past few decades, soil at high elevation has attracted more attention due to its potential impact on vegetation distribution along the elevational gradient [26–29]. Thus, the site factors need to be considered when predicting species distributions, particularly in the topographically heterogeneous regions [30] such as the cold desert ecosystem in the north-western Himalayas. In the past, a number of studies have already examined the relationship between population dynamics and environmental factors in the cold desert ecosystem [31–35], but the influence of soil physico-chemical properties on the vegetation dynamics remains largely unexplored and overlooked by researchers due to remoteness, danger, scarcity of local infrastructure, insufficient field observations and large spatial heterogeneity [36–38]. However, soil microfauna are the key components of the cold desert ecosystems and the major consumers of microbial communities [39–41]. These are the only organisms capable of consuming such biological crusts, as the higher animals are not able to survive in such harsh conditions with sufficient abundance [42]. The typical groups of soil fauna, such as springtails or oribatida, are restricted by desertic conditions; hence, the majority of feeding traits are derived mainly from those of nematodes and rotifers. In addition, Acidobacteria, Proteobacteria, Actinobacteria, Bacteroidetes, Cyanobacteria and Gemmatimonadetes are among the highly abundant taxonomic groups present in the cold desert of north-western Himalaya [43–49].

Further, the mountain ecosystems serve as natural laboratories for investigating how the environment affects the distribution of plant species, since they offer complex environmental conditions [50] due to steep the elevation gradient, which is perhaps the most fundamental and distinctive characteristic of a mountain climate [51,52]. Thus, the elevational gradient presents a complex, multi-factor gradient that affects the plant species distribution in multiple ways [53] due to hydrothermal variations [54–56]. Simultaneously, the elevation gradient also represents the influence of soil physico-chemical properties on plant species diversity and distribution [29] due to small-scale differences in soil properties with elevational gradient [57,58]. For instance, the Himalayan cold desert ecosystems in the alpine and sub-alpine zones have a coarse textured and deserted soil type with poor water and nutrient-holding capacity [59–61] and significant variations in soil properties along the elevational gradients [59], which limit the vegetation distribution.

*Juniperus macropoda* Boisser, (Syn. *J. excelsa* M, Bieb.) commonly known as the Himalayan pencil cedar, is frequently found in the dry river valleys of temperate and alpine regions of the western Himalayas at an elevation of 2400–4300 m asl [62], possessing a silty clay loam to sandy loam soil texture, which is slightly alkaline in its reaction [63]. This species is ecologically and economically significant as it meets the timber, fuel wood and fodder-related needs of the local communities [64]. Additionally, it also prevents soil erosion and enhances underground water table recharge. *J. macropoda* is a cold-resistant species that can withstand extremely harsh abiotic environmental conditions (shallow as well as stony soils, including cold, hot and dry climates) and biotic pressure [65–67]. This species is reported to have arbuscular mycorrhizal (AM) association in its root system, of which *Gigaspora albida* N.C. Schenck & G.S. Sm., *Glomus mosseae* T.H. Nicolson & Gerdemann and *Glomus fasciculatum* (Thaxt.) Gerd. & Trappe are the major arbuscular mycorrhizae [68]. These are the key biological tools for balancing soil nutrients, nutrient loss and sustainability in the cold desert ecosystem [69,70]. Additionally, they can facilitate the reclamation and revegetation of wastelands due to their potential for increasing growth, survival and biomass production under conditions of environmental stress [71,72].

However, human interferences, slow growth rate and poor natural regeneration have resulted in the degradation of natural sites the world over [73–75]. Further, since the species is under the least concerned category of the IUCN red data list, the significant decline in the mature individual due to the continuous threat from anthropogenic activities is of serious concern [76]. Additionally, the natural regeneration of juniper trees is very minimal and challenging due to grazing by animals and felling for firewood by villagers [77–79]. Therefore, juniper forests need well-planned management programs to facilitate regeneration and restoration of pure and mixed stands. Although natural regeneration and stand dynamics of the juniper forests have been studied well in Baluchistan (Pakistan), [65,80,81] the Indian Himalayas still remain unexplored in this regard. Despite the overwhelming ecological and economic importance of juniper, little emphasis has been given to studying the population dynamics and biomass production potential in relation to the soil physico-chemical characteristics in the cold desert forest ecosystem of the north-western Himalayas.

Therefore, the present study was undertaken to ascertain whether or not the soil physico-chemical properties influence the population dynamics of the *J. macropoda* along the elevation gradient in the cold desert of the north-western Himalayas. Considering that establishing a relationship between the soil physico-chemical characteristics and population dynamics along elevation gradients can improve our understanding of the spatial variation in nutrients and their impact on *J. macropoda* distribution in mountain ecosystems, this will enable us to better predict the mountain ecosystem responses to global climate change [82,83].

## 2. Materials and Methods

### 2.1. Study Area and Sampling Procedure

The study area comprises government-owned protected forest on south-facing slopes located in Lahaul Valley in the north-western Himalayan state of Himachal Pradesh, India; it spans from 32°34′ N to 32°43′ N and 77°0′ E to 77°12′ E and has an elevation range of 2500–5000 m asl (Figure 1). Lahaul Valley has an extremely harsh climate, with precipitation mainly in the form of snowfall (466.2–693.2 mm) and scanty rainfall (241.5–272.4 mm annum$^{-1}$). In July, the temperature reaches a maximum of 27.8 °C and a minimum of −13.1 °C during January [33]. Lithologically, Lahaul–Spiti contains two significant groups, viz., schistose and calcareous groups. The soil texture of the region is silty clay loam to sandy loam and alkaline along the elevation gradient [63], with an optimum status of calcium, whereas the soil thickness ranges from 0.10 to 1.0 m, with a large amount of gravel. The vegetation of the area has a relatively simple population structure, mainly comprising the individuals of one species. The top layer consists of *J. macropoda* trees, while in the lower strata, the main plant species found are *Ephedra* vulgaris L., *Juniperus communis* L., *Lonicera* spp. and *Juniperus recurva* Buch.-Ham. ex D.Don, etc.

Initially, a reconnaissance survey was conducted along the elevational gradient in the *J. macropoda* forest with the help of State Forest Department officials. Thereafter, to assess the distribution pattern and accessibility, a systematic sampling with random start was carried out. Beginning with 3000 m asl elevation as the baseline, eight elevation ranges—i.e., $E_1$ (3000–3180 m asl), $E_2$ (3180–3360 m asl), $E_3$ (3360–3540 m asl), $E_4$ (3540–3720 m asl), $E_5$ (3720–3900 m asl), $E_6$ (3900–4080 m asl), $E_7$ (4080–4260 m asl) and $E_8$ (>4260 m asl), with the elevation distance of 180 m (based on the lapse rate for the inner Himalayas)—at three separate locations were considered. At each elevation range, three plots of size 0.1 ha (31.62 × 31.62 m$^2$) were randomly established to examine the tree density and composition (24 elevation stands × 3 plots each = 72 plots). However, for regeneration survey, 2 m × 2 m plots [84] were laid out at four corners and one in the center of the main plot to record the number of seedlings (<0.5 m) and saplings (0.5–2 m) of *J. macropoda*. Trees, saplings and seedlings within each plot were counted to determine their density. The seed bank was also studied in a 1 m × 1 m plot [85] within each regeneration plot. The soil in the plot was dug up to a depth of 5 cm, followed by extracting and weighing of the seeds present in it.

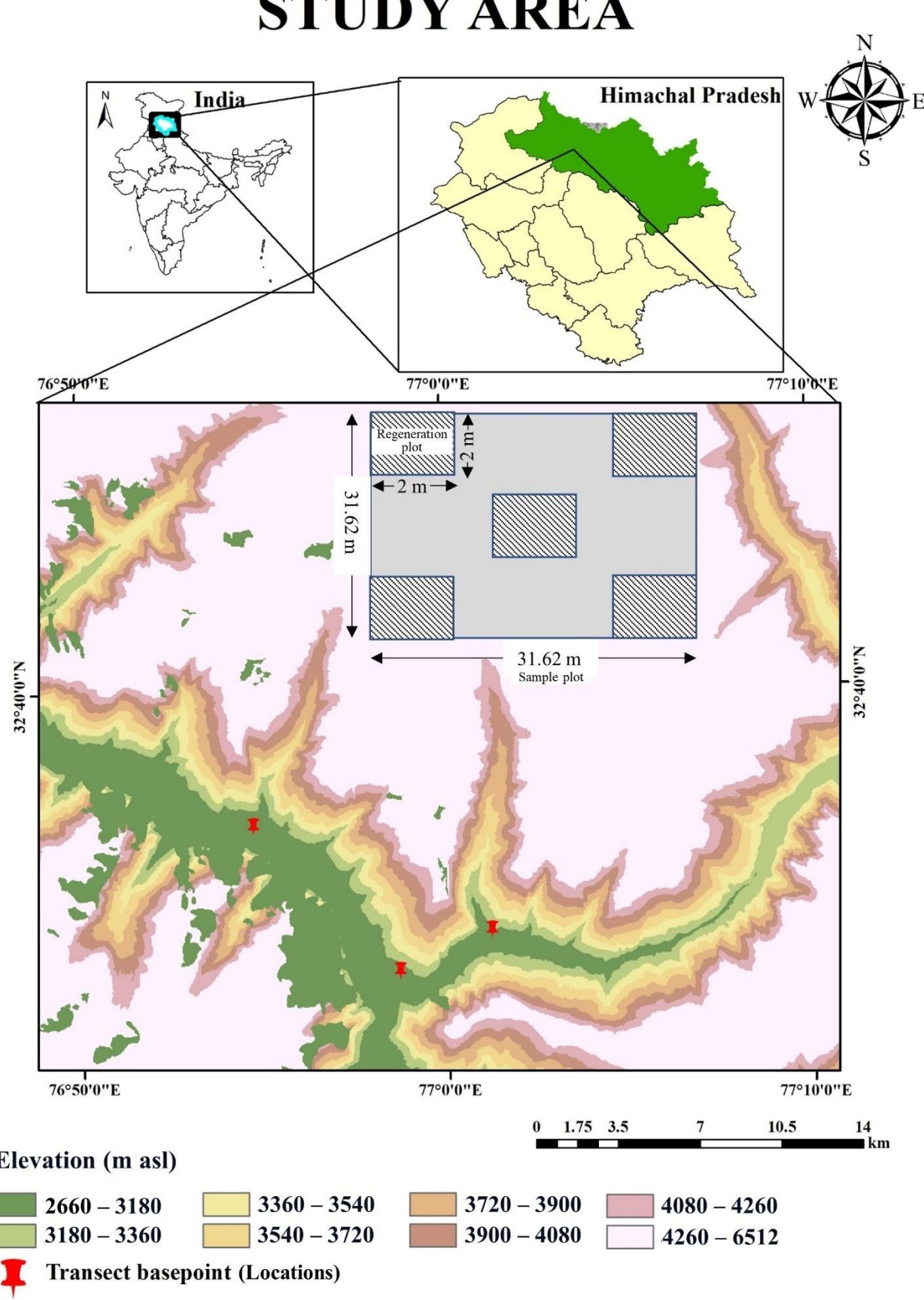

**Figure 1.** Map of the study area.

Standard measurement procedures were followed for taking primary observations such as tree height, diameter at breast height (DBH), crown height and crown diameter. The tree height was measured with a Ravi altimeter (Blume-Leiss Hypsometer). The diameter was measured by taking two measurements of stems (major and minor axis) at breast height (1.37 m) with tree callipers, and their mean was calculated as the DBH of a tree. The stem biomass (Mg ha$^{-1}$) was calculated by multiplying the specific gravity of the stem wood [86] by stem volume, i.e., stem biomass = average specific gravity of stem wood × stem volume. In the absence of an equation for estimating foliage and below-ground biomass, the equation formulated by Kramer [87] for *Juniperus occidentalis* was used. The total tree biomass was then determined by summing the above-ground and below-ground biomass. The total biomass was calculated by summing the biomass of each tree present in a particular sample plot. The vegetation carbon (Mg ha$^{-1}$) was determined as per the procedure followed by Bhardwaj et al. [88,89].

*2.2. Soil Physico-Chemical Analysis*

A composite of three soil samples were collected from each main plot (30 cm depth) with a core sampler. Bulk density was calculated as the ratio of the oven-dried soil weight to the volume of core sampler. The samples were air-dried and mixed well before they were grounded with mortars and pestles and sieved with a 2 mm mesh sieve before analysis. The samples were prepared to analyze the soil pH (method:1:2.5 soil: water suspension, digital pH meter, Jackson, [90]), electrical conductivity (method:1:2.5 soil suspension, with the help of Systronics conductivity meter), soil organic carbon (SOC) (Walkley and Black, [91] method), available nitrogen (alkaline potassium permanganate method of Subbiah and Asija, [92]), available phosphorus [93], available potassium (flame photometer method of Merwin and Peach, [94]), calcium carbonate (rapid titration method or Puri's method) and soil texture (hydrometer method). Soil carbon density (Mg ha$^{-1}$) was calculated with the method given by Nelson and Sommers [95], viz., soil carbon density = soil bulk density (g cm$^{-3}$) × soil depth (cm) × SOC. From this, the total carbon density (Mg ha$^{-1}$) was calculated by adding the soil carbon density (Mg ha$^{-1}$) and vegetation carbon density (Mg ha$^{-1}$).

*2.3. Statistical Analysis*

The data obtained were subjected to analysis of variance technique [96] for comparing the difference among the elevational ranges with respect to the characteristics under study. A multiple comparison test, i.e., the least significant difference (lsd), was performed for pair wise comparison of elevational ranges. The statistical analysis was performed in R software using the following function:

*anova<-lm(dep_var ~ ind_var)*
*summary (anova)*
*lsd<-LSD. Test (anova, "ind_var")*

Further, the correlation and regression analyses were performed to establish a relationship between the physico-chemical characteristics and the population dynamics. In order to study how population dynamics are being influenced by the mentioned characteristics, multiple linear regressions were employed as simple regression analysis, considering that only one variable as regressor does not serve the purpose of drawing the overall conclusions since it ignores the existing multiple correlation present among the characteristics. R software functions used were:

*Model<-lm (dep_vr ~ ind_ vrs, data = data)*
*ols_step_both_p(model)*

Further, the fitting of model was judged by adjusted $R^2$ and standard errors of the estimates of regression coefficients. Variance inflation factors (VIF) [97] are also an important guide to the validity of the model. R function for computation of VIF of each regressors is:

*Library (car)*
*Vif (Model)*

If VIF exceeds 5, that particular coefficient is poorly estimated or unstable because of near-linear dependences among the regressors [98].

## 3. Results

### 3.1. Stand Characteristics and Population Dynamics at Tree Line

The results revealed that the elevation gradient had a significant influence on the growth and development traits of *J. macropoda*. In general, all the traits—viz., tree density (141.67–683.33 N ha$^{-1}$), average height (2.90–6.43 m), diameter (DBH) (8.03–14.80 cm), basal area (0.73–1297 m$^2$ ha$^{-1}$), volume, crown height (0.23–0.53 m), crown width (1.53–2.50 m) and crown length (2.67–5.90 m)—showed a declining trend from $E_1$ to $E_6$, with maximum values at the mid of the elevation (Figure 2). Furthermore, the elevation gradient was found to significantly influence the distribution of the seedling, saplings, their height and diameter growth, seed storage (m$^{-2}$) and seed weight (g), respectively (Table 1).

The first three elevation ranges did not depict a definite trend for the aforementioned traits. However, the tree density was recorded highest (683.33 N ha$^{-1}$) at $E_1$ (3000–3180 m) and showed a sharp decline after $E_3$ (3360–3540 m asl) elevation range up to $E_6$ (141.67 N ha$^{-1}$), while the seedling density initially increased with the elevation gradient from $E_1$ (166.67 N ha$^{-1}$) up to $E_4$ (3540–3720 m) (433.33 N ha$^{-1}$) and, thereafter, a significant decline was recorded. Likewise, an initial rise in the sapling density was recorded up to $E_3$ (3360–3540 m asl) (533.33 N ha$^{-1}$), after which it ultimately declined, with few rises and falls till $E_7$ (133.33 N ha$^{-1}$). The average height and diameter growth of the seedlings, as well as the saplings at different elevation ranges, were statistically similar, except those at $E_7$ (4080–4260 m asl), which showed minimum height (seedling—0.22 m, sapling—0.48 m) and diameter (seedling—0.21 cm, sapling—2.00 cm) growth. The maximum seed storage of *J. macropoda* (64.00 number of seeds m$^{-2}$ and 3.00 g seed weight) occurred at an elevation of $E_3$ (3360–3540 m asl) and showed a steady declining trend before and afterward. Despite the absence of vegetation at the highest level of the elevation range (>4260 m), a fractional number of seeds was recorded. Moreover, *J. macropoda* forests at the tree line formed a reverse J- shaped structure, with a maximum distribution of individuals in 0–10 cm diameter class, followed by 10–20 cm and 20–30 cm, respectively (Figure 3). Numerous seedlings/saplings became established at the tree line and above the upper limit (3900–4080 m asl) of contiguous forests. The uppermost individual of *J. macropoda* was found at 4300 m asl, well above the current tree line.

### 3.2. Biomass and Carbon Density (Mg ha$^{-1}$)

The biomass and carbon density of the *J. macropoda* forest under present investigation were significantly influenced by the elevational gradient (Figure 4). The maximum total biomass (111.51 Mg ha$^{-1}$) and carbon density (144.6 Mg C ha$^{-1}$) were recorded at $E_1$ (3000–3180 m asl). However, the values remained statistically identical for the first three elevation ranges. Likewise, the last three elevational ranges, viz., $E_6$ (3900–4080 m asl), $E_7$ (4080–4260 m asl) and $E_8$ (>4260 m asl), were also observed to be at par for biomass and carbon density, except for soil carbon density. The above-ground biomass, below-ground biomass, total biomass and plant carbon density declined uniformly with the increase in elevation. Moreover, the plant carbon density at the upper elevational ranges, viz., $E_6$ (3900–400 m asl), $E_7$ (4080–4260 m asl) and $E_8$ (>4260 m asl), showed much less biomass density (6.16 Mg ha$^{-1}$, 2.04 Mg ha$^{-1}$ and 2.01 Mg ha$^{-1}$) as compared to that at the lower elevation ranges. The soil carbon density (SOC) attained its peak (89.13 Mg C ha$^{-1}$) at $E_3$ (3360–3540 m) and displayed a declining trend thereafter.

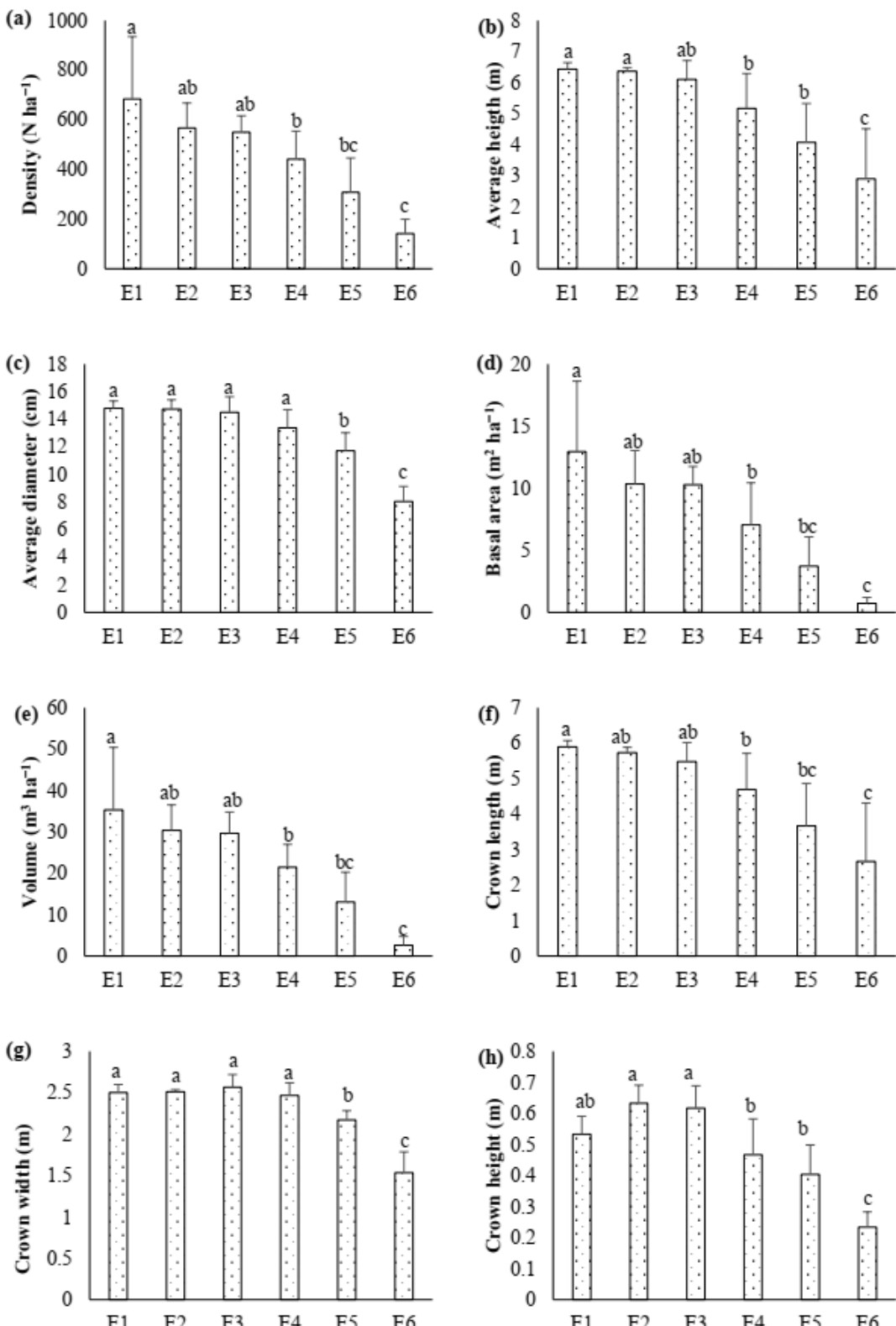

**Figure 2.** Growth and developmental characteristics of *J. macropoda* at different elevation ranges in north-western Himalaya where: (**a**) denotes density (N ha$^{-1}$), (**b**) average height (m), (**c**) average diameter (cm), (**d**) basal area (m$^2$ ha$^{-1}$), (**e**) volume (m$^3$ ha$^{-1}$), (**f**) crown length (m), (**g**) crown width (m) and (**h**) crown height (m). The error bar shows the mean standard deviation. Different letters ($^{abc}$) indicate significant differences at $p < 0.05$.

**Table 1.** Regeneration status and seed bank at different elevational ranges in cold desert of north-western Himalaya.

| Elevation (m asl) | Seedling (<0.5 m) | | | Sapling (0.5–2 m) | | | Seed Bank | |
|---|---|---|---|---|---|---|---|---|
| | Density (N ha$^{-1}$) | Height (m) | Diameter (cm) | Density (N ha$^{-1}$) | Height (m) | Diameter (cm) | Number of Seeds m$^{-2}$ | Seed Weight (g m$^{-2}$) |
| E$_1$ (3000–3180 m) | 166.67 [bcd] | 0.33 [a] | 0.52 [a] | 333.33 [b] | 1.57 [a] | 3.39 [a] | 58.67 [ab] | 2.78 [a] |
| E$_2$ (3180–3360 m) | 233.33 [bc] | 0.40 [a] | 0.77 [a] | 500.00 [ab] | 1.62 [a] | 3.38 [a] | 44.67 [b] | 2.13 [ab] |
| E$_3$ (3360–3540 m) | 266.67 [b] | 0.34 [a] | 0.59 [a] | 533.33 [a] | 1.55 [a] | 3.41 [a] | 64.00 [a] | 3.00 [a] |
| E$_4$ (3540–3720 m) | 433.33 [a] | 0.37 [a] | 0.53 [a] | 300.00 [b] | 1.51 [a] | 3.06 [a] | 51.33 [ab] | 2.47 [ab] |
| E$_5$ (3720–3900 m) | 200.00 [bcd] | 0.36 [a] | 0.55 [a] | 366.67 [b] | 1.48 [a] | 3.14 [a] | 35.67 [bc] | 1.6 [b] |
| E$_6$ (3900–4080 m) | 133.33 [cd] | 0.35 [a] | 0.64 [a] | 266.67 [b] | 1.45 [a] | 3.05 [a] | 19.00 [c] | 0.9 [bc] |
| E$_7$ (4080–4260 m) | 100.00 [cd] | 0.22 [b] | 0.21 [b] | 133.33 [c] | 0.48 [b] | 2.00 [b] | 3.67 [c] | 0.2 [c] |
| E$_8$ (>4260 m) | - | - | - | - | - | - | 0.2 [d] | 0.003 |
| SE (m) | 42.10 | 0.043 | 0.09 | 52.20 | 0.13 | 0.39 | 5.51 | 0.28 |
| Lsd | 131.16 | 0.13 | 0.30 | 162.63 | 0.44 | 1.13 | 18.44 | 0.94 |
| *p*-value | 0.00240 | 0.01600 | 0.04469 | 0.00233 | 0.00121 | 0.023185 | 0.00010 | 0.00021 |

Values within columns represented by same alphabetical letter are statistically on par, and those represented by different letters are significant, at 5% level of significance. SE(m): standard error of mean; *p*-value: probability of obtaining the observed or more extreme results when null hypothesis is true; lsd: least significant difference or critical value at 5% level of significance.

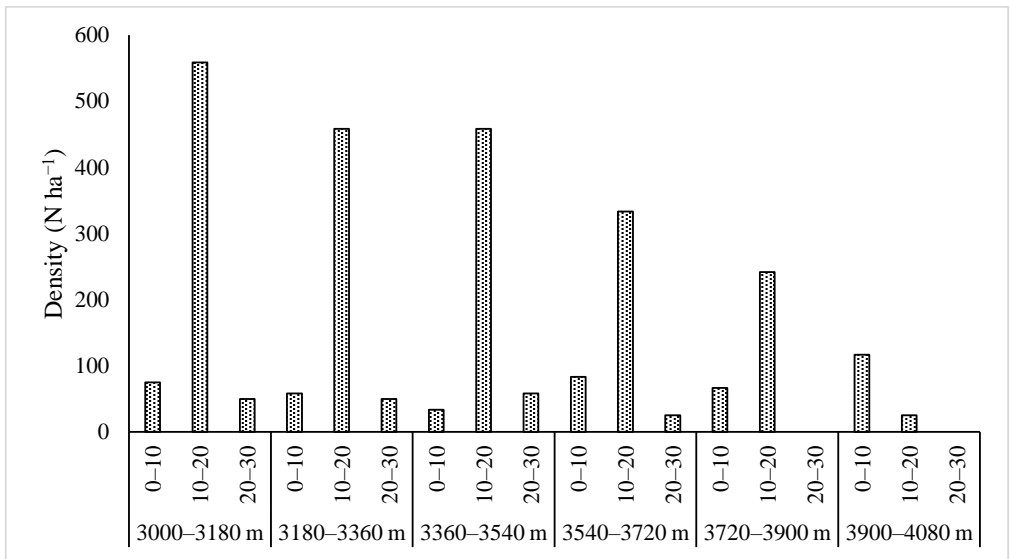

**Figure 3.** Distribution of *J. macropoda* in cold desert of Lahaul Valley along elevation and size classes.

### 3.3. Soil Physico-Chemical Properties

The soil physico-chemical characteristics of the study area varied significantly along the elevation gradient (Table 2). The soil pH and EC were significantly influenced by the elevational gradient. However, the values showed fluctuation until the mid of the elevation E$_5$ (3720–3900 m asl) and, overall, declined with the elevation gradient. The elevation E$_1$ (3000–3180 m asl) recorded the maximum pH of 6.70, whereas the maximum EC (0.69 dSm$^{-2}$) was observed at E$_4$ (35,400–3720 m asl), E$_5$ (3720–3900 m asl) and E$_6$ (3900–4080 m asl). A wide variation was seen for bulk density, organic carbon (%), and CaCO$_3$ (%) along the elevational gradient. The bulk density declined with an increase in the elevation and was found to be maximum (1.33 g cm$^{-3}$) at E$_8$ (>4260 m). OC (%) was profoundly influenced by elevational effects and was maximum (2.86%) at E$_3$ (3360–3540 m asl) elevational range. Likewise, the highest CaCO$_3$ content was recorded at the elevational range E$_2$ (3180–3360 m asl) (5.42%), which remained statistically identical to that at E$_1$ (3000–3180 m asl) and E$_3$ (3360–3540 m asl).

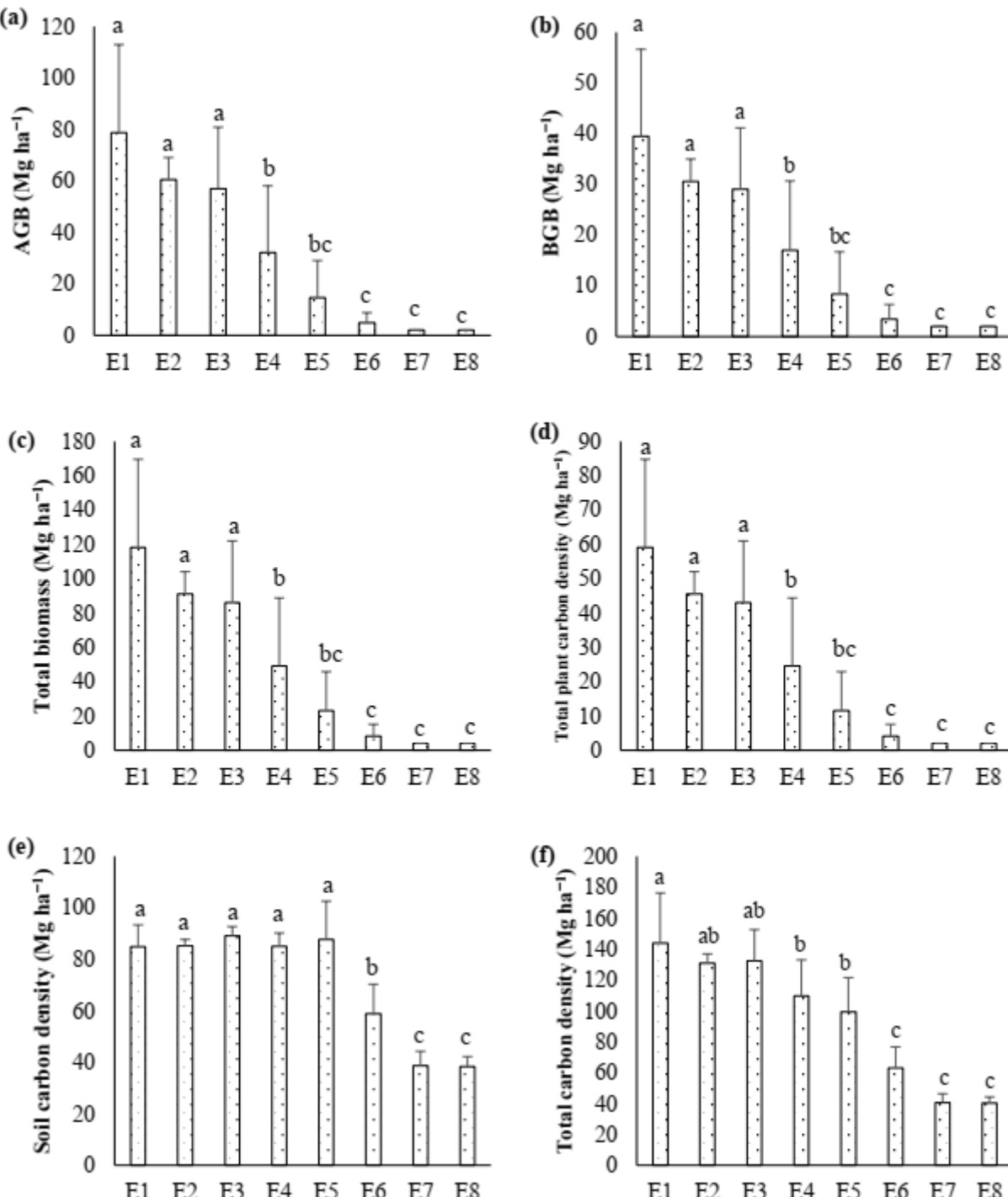

**Figure 4.** Biomass and carbon density (Mg ha$^{-1}$) at different elevations in the cold desert of north-western Himalaya where: (**a**) above-ground biomass (AGB) (Mg ha$^{-1}$), (**b**) below-ground biomass (BGB) (Mg ha$^{-1}$), (**c**) total biomass (Mg ha$^{-1}$), (**d**) total plant carbon density (Mg C ha$^{-1}$), (**e**) soil carbon density (Mg C ha$^{-1}$) and (**f**) total carbon density (Mg C ha$^{-1}$). The error bar shows the mean standard deviation. Different letters ($^{abc}$) indicate significant differences at $p < 0.05$.

**Table 2.** Soil physico-chemical characteristics at a different elevation in the cold desert of north-western Himalaya.

| Elevation (m asl) | Soil pH | EC (dS m$^{-2}$) | Bulk Density (g cm$^{-3}$) | SOC (%) | CaCO$_3$ (%) | Soil Texture | | | Available N (kg ha$^{-1}$) | Available P (kg ha$^{-1}$) | Available K (kg ha$^{-1}$) |
|---|---|---|---|---|---|---|---|---|---|---|---|
| | | | | | | Sand (%) | Silt (%) | Clay (%) | | | |
| E$_1$ (3000–3180 m) | 6.70 [a] | 0.67 [a] | 1.01 [b] | 2.80 [a] | 5.17 [ab] | 33.97 [c] | 31.23 [ab] | 30.37 [ab] | 342.30 [a] | 6.90 [ab] | 235.94 [a] |
| E$_2$ (3180–3360 m) | 6.55 [ab] | 0.65 [ab] | 1.03 [b] | 2.77 [a] | 5.42 [a] | 32.70 [cd] | 32.07 [a] | 29.57 [b] | 338.41 [a] | 7.48 [a] | 283.87 [a] |
| E$_3$ (3360–3540 m) | 6.55 [ab] | 0.60 [abc] | 1.04 [b] | 2.86 [a] | 5.21 [ab] | 31.57 [d] | 31.97 [a] | 30.90 [a] | 348.18 [a] | 7.69 [a] | 276.69 [a] |
| E$_4$ (3540–3720 m) | 6.33 [b] | 0.69 [a] | 1.06 [b] | 2.69 [a] | 4.58 [bc] | 33.13 [c] | 31.07 [ab] | 29.00 [bc] | 334.63 [a] | 8.67 [a] | 278.37 [a] |
| E$_5$ (3720–3900 m) | 6.54 [ab] | 0.69 [a] | 1.27 [ab] | 2.30 [b] | 4.05 [c] | 33.20 [c] | 30.73 [b] | 27.87 [c] | 277.03 [b] | 5.73 [ab] | 173.79 [b] |
| E$_6$ (3900–4080 m) | 6.60 [a] | 0.69 [a] | 1.06 [b] | 1.84 [c] | 3.72 [c] | 37.63 [b] | 30.37 [b] | 27.57 [c] | 218.51 [c] | 5.68 [ab] | 169.88 [b] |
| E$_7$ (4080–4260 m) | 6.35 [b] | 0.56 [b] | 1.21 [ab] | 1.08 [d] | 3.75 [c] | 38.23 [b] | 30.57 [b] | 24.97 [d] | 189.32 [d] | 4.14 [b] | 147.19 [b] |
| E$_8$ (>4260 m) | 6.33 [b] | 0.53 [b] | 1.33 [a] | 0.96 [d] | 3.79 [c] | 40.97 [a] | 30.23 [b] | 23.07 [e] | 136.08 [e] | 3.66 [b] | 127.53 [b] |
| SE (m) | 0.08 | 0.03 | 0.07 | 0.08 | 0.22 | 0.45 | 0.37 | 0.42 | 7.37 | 0.99 | 29.42 |
| Lsd | 0.24 | 0.10 | 0.21 | 0.23 | 0.67 | 1.38 | 1.12 | 1.27 | 22.56 | 3.03 | 90.09 |
| *p*-value | 0.02897 | 0.01685 | 0.02358 | 0.000 | 0.00010 | 0.0000 | 0.01977 | 0.0000 | 0.0000 | 0.03159 | 0.00981 |

Values within the columns are represented by same alphabetical letter are statistically on par, and those represented by different letters are significant, at 5% level of significance. SE(m): standard error of mean; *p*-value: probability of obtaining the observed or more extreme results when null hypothesis is true; lsd: least significant difference or critical value at 5% level of significance.

The sand (%), silt (%) and clay content (%) varied significantly along the elevational gradient. The minimum sand content was recorded at E$_3$ (3360–3540 m asl) and showed a marked increase thereafter until the last elevational range of E$_8$ (>4260 m asl), where the maximum sand content recorded was 40.97%. The maximum silt content (32.07%) was reported at E$_2$ (3180–3360 m asl) elevational range. The clay content (30.90%) was maximum at A$_3$ (3360–3540 m asl), and thereafter, it showed a declining trend. Based on the sand, silt and clay content, the textural class of *J. macropoda* was classified as sandy loam. The elevational gradients markedly affected the levels of available N, P and K (kg ha$^{-1}$) of the soil. The maximum values of available N (348.18 kg ha$^{-1}$), P (8.67 kg ha$^{-1}$) and K (283.87 kg ha$^{-1}$) were recorded at E$_3$ (3360–3540 m asl), E$_4$ (35,400–3720 m asl) and E$_2$ (3180–3360 m asl), respectively.

### 3.4. Correlation and Regression Studies

Evaluation of the significance of the elevation gradient and soil characteristics for the distribution of the tree, seedling and saplings of *J. macropoda* was carried out on the basis 24 plots on which soil variables were recorded in the soil layer.

The available N, available K, EC, SOC (%), SCD, clay and sand bore a significant relationship with the seedlings distribution at the tree line (Figure 5). In addition to the above soil factors, saplings also required available P, CaCO$_3$ and silt contents for their establishment. Furthermore, trees also bore a significant positive relation with soil pH. The distribution of seedlings, saplings as well as trees bore a strong negative relationship with sand content. In general, the intensity of the correlation coefficient(r) of the above mentioned physico-chemical characteristics followed the order: seedling < sapling < tree.

Further, an attempt was made to establish a relation between the seedling, sapling and tree density (N ha$^{-1}$) with the studied soil parameters. Multiple linear regression was performed and variance inflation factors (VIF) were computed for each regressor (Table 3). These VIF values indicated the multicollinearity among the regressors that yielded a non-significant regression coefficient. Therefore, stepwise regression analysis was performed to extract the set of regressors that significantly affected the response variables. In stepwise regression analysis, the variance inflation factors for each selected regressor were less than 5, which indicated the absence of multicollinearity among the regressors, thus making the least square estimate of the parameters valid. The seedling density could be predicted (Adj R$^2$ = 0.686) with the help of available N, CaCO$_3$, pH, available P and silt content, while sapling density showed the best prediction with available P and sand contents (Table 3). In comparison to the seedling and sapling density, the tree density could be better predicted with the equation: $-2478.323 + 439.175$ SOC $+ 276.616$ pH (Adj R$^2$ = 0.837). The above-ground biomass was best predicted (R$^2$ = 0.709) using CaCO$_3$ and SOC contents, and SOC and BD predicted SCD with a very high degree of confidence (R$^2$ = 0.980).

| Soil parameters | Seedling Density | Sapling Density | Tree Density | r |
|---|---|---|---|---|
| Available N | 0.629* | 0.815* | 0.896* | −1 |
| Available P | 0.362 | 0.686* | 0.527* | |
| Available K | 0.451* | 0.700* | 0.650* | |
| SOC (%) | 0.601* | 0.795* | 0.918* | |
| SCD | 0.569* | 0.756* | 0.856* | |
| CaCO$_3$ (%) | 0.237 | 0.631* | 0.792* | |
| pH | −0.067 | 0.306 | 0.529* | 0 |
| EC | 0.423* | 0.307 | 0.380 | |
| PD | −0.116 | 0.400 | 0.358 | |
| BD | −0.344 | −0.525* | −0.536* | |
| Sand (%) | −0.584* | −0.824* | −0.821* | |
| Silt (%) | 0.273 | 0.583* | 0.638* | |
| Clay (%) | 0.534* | 0.796* | 0.856* | +1 |

**Figure 5.** Simple correlations between seedling/sapling/tree abundance of *J. macropoda* with physico-chemical characteristics in north-western Himalayas. * Significant at 5% level of significance. SOC—soil organic carbon; SCD—soil carbon density; EC—electrical conductivity; PD—particle density; BD- bulk density.

**Table 3.** Regression analysis between seedling, sapling, tree density (N ha$^{-1}$), above-ground biomass and soil carbon density and the soil characteristics.

| Dependent Variable | Equation | Adj R$^2$ | *p*-Values of Estimates | VIF |
|---|---|---|---|---|
| Seedling density | 480.414 + 2.412 Available N—158.297 CaCO$_3$—273.412 pH—15.255 Available P + 52.915 Silt | 0.686 | <0.05, 0.091, 0.012, 0.001, 0.099 | 4.01, 4.18, 1.46, 2.01, 3.02 |
| Sapling density | 1241.549 + 21.853 Available P—30.072 Sand | 0.614 | 0.04, 0.004 | 1.51,1.51 |
| Tree density | −2478.323 + 439.175 SOC + 276.616 Ph | 0.776 | <0.05, 0.04 | 1.01, 1.01 |
| Above-ground biomass | −103.536 + 22.637CaCO$_3$ + 15.749 SOC | 0.709 | 0.04, 0.01 | 2.30,2.30 |
| Soil carbon density | −47.280 + 32.460 SOC + 42.610 BD | 0.988 | <0.05, <0.05 | 1.59,1.59 |

## 4. Discussion

### 4.1. Population Dynamics, Biomass and Carbon Density

In the mountainous forest ecosystem, the diverse topography and elevation range resulted in an impressive effect on the environmental microclimate, vegetation cover and soil properties [99,100] over a short distance, often with sharp transitions (ecotone) in the vegetation sequences [101,102]. Particularly, the environmental factors, such as temperature, light intensity, soil moisture and fertility status in the alpine ecosystem are the most decisive factors and may explain the decrease in the growth and developmental traits along an elevation gradient [61,103,104].

In the present investigation on *J. macropoda* forest in a cold desert ecosystem, the average number of seedlings, saplings and trees recorded were 219 N ha$^{-1}$, 347 N ha$^{-1}$

and 448 N ha$^{-1}$, respectively. The previous studies revealed a great deal of variation in *J. macropoda* forests, and the density of juniper tract ranged from 56–332 ha$^{-1}$ [80] to 29–268 ha$^{-1}$, with an average of 176 ± 77 individuals ha$^{-1}$ [105]. Conversely, Atta [106] related the reduced juniper stock density to the biotic pressure. In pure stands of juniper on dry south-western aspects, the tree density was found to be greater than 15.33 individuals per 100 m$^2$ [107]. The higher tree density in our study may be attributed to effective conservation laws, which protect the species legally from anthropogenic disturbances.

Furthermore, the tree growth and development traits, i.e., crown height, crown width, crown length and stand characteristics, of *J. macropoda* were highest at lower elevation ranges and declined significantly along the elevation gradient. Similarly, Momeni et al. [108] stated that besides climatic factors, the elevation gradient and soil properties had a substantial effect on the tree density, regeneration, stand basal area, slenderness ratio and crown diameter of *J. macropoda*. The decline in the tree growth and development traits may also be attributed to light intensity, which is the most important ecological resource for photosynthesis, as higher elevations have higher light intensity than lower ones, which affects the tree stand growth [109–114]. Therefore, the trees growing at low elevations generally had greater stem heights to enable them to capture more light owing to the vertical growth [115,116] whereas the trees growing at higher elevations produced thicker stems that grew radially, as the temperature is colder there. The maximum height and diameter of the J. trees in our study were 6.43 m and 14.80 cm, respectively. Similarly, Abido and Kurbaisa [117] reported that in Iran junipers reached a height of 6.5 m, though with a diameter of approximately 80 cm. At the study sites, the production and spread of viable seeds seemed sufficient to generate relatively high rates of seedlings, even beyond the actual upper limit of continuous forests (3720–4080 m asl).

In the current study, *J. macropoda* trees formed a timber line up to an elevation range of 3900–4080 m asl, while the stunted growth of seedlings and saplings was reported up to an elevation range of 4080–4260 m asl. A fractional quantity of in situ seed storage was also observed beyond the tree line, but with no recruits. Seedlings and saplings showed their peak value at mid-elevation, while the density of trees declined gradually from the first elevation range onward. Moreover, the population structure, i.e., distribution of seedlings, saplings and trees along the elevation gradient, followed the fertility gradient. In comparison to other elevation ranges, elevations A$_2$ (3180–3360 m asl), A$_3$ (3360–3540 m asl) and A$_4$ (3540–3720 m asl) had higher vegetation and regeneration and the highest concentration of most soil nutrients. This may be due to the local topographic and site factors favoring the vegetation of *J. macropoda*. Likewise, the status of the primary nutrients, i.e., available N, P and K, decreased significantly along the elevation gradient, which indicates that besides climatic variables, soil nutrients play a pivotal role in controlling the distribution of vegetation and regeneration of *J. macropoda* forest along the elevation gradient. It could be speculated that the nutrient deficiency, in addition to the cold environment, limited the extension of juniper beyond the tree line [118,119], which was thus restricted to an elevation of 4260 m asl in our present study. As the nutrient availability is constrained by the low temperature, which controls the tree lines at high elevations [25], the levels of nutrient availability simultaneously decreased with an increase in the elevation [120].

The biomass density (above and below, as well as total) and carbon storage in the present study area were substantial mainly up to an elevational range of 3720–3900 m asl and were very less thereafter. The soil carbon density maintained a declining trend from the middle to upper elevation ranges. At each elevation range, the soil carbon density stored in *J. macropoda* forest was approximately two times the biomass density. The maximum biomass carbon density (55.76 Mg C ha$^{-1}$) was recorded at the lowermost elevation range, which was quite lower than the values reported by Norris et al. [121] for *J. virginiana* in the tall grass prairies of a North American forest. A linear negative relationship between biomass and elevation was expected due to declining temperature and nutrients with elevation [122]. Moreover, a higher soil carbon density at lower and middle elevation ranges may be attributed to the combined effect of vegetation biomass, low temperature

and local topography, which reduces the soil erosion. Generally, higher precipitation, in the form of snow, at high altitudes than in lower altitudes and subzero temperatures caused the hyperaridity of the soil at high altitudes and suppression of microbial and enzymatic activities, which resulted in the least soil organic matter decomposition that caused the higher accumulation of SOC at higher elevation [123–125]. Conversely, in our study, the soil organic carbon decreased with the increase in elevation, which may be attributed to the fact that the presence of more rocks with sandy texture soils at the higher elevation originated from parent material (schistose and calcareous group) with intense solar radiation, which inhibited the presence of vegetation, thus making the higher elevations cold, dry and devoid of soil nutrients and moisture. Thus, soils at the cold desert high elevations are coarse-textured, permeable and uninhabited, and their poor water- and nutrient-holding capacity has resulted in low nutrient availability for growing plants [49–51]. Hence, the subalpine and alpine ecosystems of Indian Himalayas have low biomass production and soil carbon density [126].

### 4.2. Soil Physico-Chemical Properties and Correlation Studies

The extreme nature of climatic and topographical conditions may affect soil properties at cold desert high elevations [125]. Climatic factors, especially the air temperature, annual precipitation and water balance, determine the direction and intensity of physical, chemical and biological processes, including weathering, in soil and on its surface [127–130]. Therefore, the soil of the present study area showed significant variability along the elevational gradient, similar to the previously published literature [59,131,132]. The decreasing trend in pH and EC with higher elevation shows that lower elevation sites have more cumulative salt accumulation. This may be due to the higher accumulation of base-forming cations and the higher accumulation of $CaCO_3$ in the lower and mid-elevation ranges than in the upper ranges. The bulk density showed a slightly increasing trend with the elevational gradient, which may be ascribed to the decreasing level of SOC (%) and increasing level of sand content with increasing elevation. Bulk density depends on the organic matter, soil structure and texture, as well as the freezing and thawing process [133,134]. The general declining trend of soil organic carbon density with increasing elevational ranges is likely due to the decreased biomass production potential with elevational advancement [135]. Asadi et al. [136] also concluded that the organic matter content of soil decreases as elevation increases. However, the high amount of SOC at the mid-elevations could be attributed to a deposition center from the upper parts of the hills. Nutrients are considered essential for maintaining the continuity and stability of all ecosystems [137]. The trends for the available N, P, K and SOC (%) contents of the soil in the study area were the same and could be directly associated with the organic content on the forest floor [138,139], along with the high sand contents leading to the deficient nutrient-holding capacity at these gradients. The variation in the available P and K along the elevational gradient could have been due to the accumulation of eroded soil, relatively abundant in P content at mid or lower elevations from the upper elevation ranges.

Climate and parent material profoundly influence the soil characteristics [113,140] as the soil of the region has a high level of $CaCO_3$. Since the calcareous soils occur naturally in the arid and semi-arid regions because of relatively little leaching [141], it is presumed that the calcareous soils of this cold desert high elevation might have been developed by predominantly available calcareous parent material (calcite mineral) throughout the soil profiles. Simultaneously, the higher concentration of $CaCO_3$ at lower elevational ranges may be due to the higher evaporation rate of the soil and high soil mineralization compared to the higher elevations ranges. The sandy loam textural class prevailed throughout the elevation gradient. Furthermore, soils of the cold desert ecosystems originated from weathered rocks; hence, they are immature with a large proportion of sand [60]. Likewise, in the present study, the sand fraction increased with elevation, whereas the silt and clay fractions occurred in reverse order, indicating the dominance of sand-forming minerals in the parent materials. The slow process of soil formation along the elevation gradients

results in a low content of clay particles and a low content of available mineral nutrients in this soil [141]. Additionally, soil microorganisms are considered to be primarily responsible for litter decomposition and soil nutrient dynamics [44,45]. The symbiotic association of soil microorganism, i.e., Acidobacteria, Proteobacteria, Actinobacteria, Bacteroidetes, Cyanobacteria and Gemmatimonadetes, boost the nutrient uptake of soil nutrients by the plant by exploration of external hyphae of the soil beyond the root hair and depletion zones. Soil microorganisms are the essential components of soil biodiversity and play a crucial role in maintaining soil processes and nutrient availability, which are critical to sustaining the functioning of cold desert ecosystems [46,47]. However, environmental factors such as water content, total soil organic carbon, total soil nitrogen, soil salinity and soil pH are vital in determining the bacterial community in cold deserts [49].

Moreover, all the components of the *J. macropoda* vegetation, i.e., seedlings, saplings and trees, were found to have a significantly negative correlation ($p < 0.05$) with sand content. Similarly, Sarangzai et al. [81] also confirmed that the vegetation characteristics of *J. excelsa* had a significant relationship with soil characteristics. The adjusted $R^2$ values of the regression equation of tree density with various physico-chemical traits were greater than those of the seedling and sapling density. Thus, trees require more of these growth factors than seedlings and saplings. Owing to the very low levels of the soil nutrients at or above >4260 m asl and a high proportion of sands, the extension of the tree line seems to have become static, although *J. macropoda* may show sporadic appearance at 4300 m asl or intensification under the influence of global warming in future.

## 5. Conclusions

The stand structure and growth characteristics of *J. macropoda* vary significantly along the elevation gradient studied, indicating that the habitat of the *J. macropoda* forests lying in the inner north-western Himalayas are very sensitive to changing environmental conditions. The study concluded that though the climatic factors may be decisive in determining the distribution of *J. macropoda* along the elevational gradient, the role of soil parameters should not be overlooked, as the soil characteristics also have a considerable influence on the distribution of seedlings, saplings, tree density and biomass of *J. macropoda* forests along the elevational gradient. Since the density of seedlings, saplings, trees and biomass production followed the soil fertility gradient with optimum growth at the mid-elevation ranges, the ecotone zone is experiencing the regeneration of new seedlings. Apart from the topography, the available N, P, K, SOC (%) and clay, as well as sand contents, proved to be significant explanatory variables of seedlings, saplings, tree species distribution and biomass production in the cold desert. In a nutshell, the seed-based regeneration within and beyond the tree line ecotones of cold desert does not appear to limit tree lines from being sensitive to climate/global warming responses. Overall, the present study helps in understanding the relationships among population distribution, biomass production and environmental factors in the cold desert forest ecosystems of the north-western Himalayan region and could enable foresters, silviculturists and environmentalists to apply these findings to forest management and vegetation restoration programs.

**Author Contributions:** Conceptualization, methodology, validation, formal analysis, investigation, D.K. and D.R.B.; data curation, writing—original draft, D.K. and D.R.B.; writing—review and editing, software visualization, P.S., B. and N.S.; writing—review and editing, supervision, N.A.-A. and N.T.T.L. All authors have read and agreed to the published version of the manuscript.

**Funding:** This research received no external funding.

**Data Availability Statement:** Data could be provided on reasonable request from the first author.

**Acknowledgments:** The authors are grateful to the Head of the Department of Silviculture and Agroforestry, Y.S. Parmar, University of Horticulture and Forestry, Solan (HP), India, for providing the necessary facilities during the study. We are also thankful to anonymous reviewers for their valuable suggestions and comments on the overall improvement of the manuscript.

**Conflicts of Interest:** The authors declare no conflict of interest.

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
