# Peer review of "Population Dynamics of Juniperus macropoda Bossier Forest Ecosystem in Relation to Soil Physico-Chemical Characteristics in the Cold Desert of North-Western Himalaya"

_forests, doi:10.3390/f13101624_

Round 1

Reviewer 1 Report

Comments :

Abstract & Introduction:

Please stated the main problem in the Abstract, Why did you study J. macropoda only? Is there any problem with regeneration and conservation? Is this species included in CITES Appendix list?  Why did you study phyico-chemical soils? As we know, soil biology and microbiology are important to study in this case. Please re-write the objective of this research more clearly.

Line 62-73 review the role of microbes in himalayas mountain ecosystem should be explained shortly.

Line 74-92 review the role of mycorrhizal fungi with J. macropoda should be elaborated

Discussion:

The study focused on Physico-chemical. Please make a discussion: one para about the role of microbes (inc. mycorrhizal fungi) associated with the trees in below ground relationship with deposit carbon in below ground, especially in Himalaya ecosystems and other hilly mountain ecosystems. There are some references that should be cited.

Author Response

First of all, we would like to thanks the editor and reviewers for their constructive comments to improve the quality of the manuscript in good ways. As per your kind instruction and comments, manuscript has been revised fruitfully and resubmitting the same for your kind perusal and necessary decision

Reviewer 2 Report

Dear Authors

Thank you for inviting me for reviewing a MS entitled Population dynamics of Juniperus macropoda Bossier forest ecosystem in relation to soil physico-chemical characteristics in the cold desert of northwestern Himalayal. This is very important contribution on population dynamics of high elevation plant in terms of soil characteristics. However it needs some revision in terms of presenting results followed by discussion. I would suggest to write the results in a more straightforward way and discussion focusing on what the results infer. Here are few points to be revised.

Line 17 ecosystem that has been

Limit key words to 6 words

Line 59 replace the term neglected

Line 65 just mention climate

Line 74-81 Mention this paragraph under materials and methods

Line 85 well-concerted

Line 125 what is the basis of 2 m × 2 m sampling unit

Line 128-129 why 1 m × 1 m for seed bank?

Line 151 which plot? Plot size?

Line 154 grounded

Line 155 analyzes

Line 322-328 write below your own results

Line 357-398 this paragraph is too long, split it into two

Line 400-409 here again shift these information below the main results of the study

Line 451 stand structure refers to the age structure of plant, this has not been worked out for this study so where does it come from?

Line 466-468 Do not write such a loose sentence in your conclusion section, write only what the results infer

I think references are quite a lot, some references could be omitted.

Author Response

(The authors gave the same response as above.)

Reviewer 3 Report

The manuscript addresses a crucial issue related to the deployment of Juniperus macropoda in the elevational and/or soil fertility gradients in NW Himalaya. The topic relates to the aims of Forests and its section Forest Ecology and Management.

Overall I am satisfied with the relevance of the presented paper. But, it needs to be improved in case of minor editing flaws and some questions that I attached below. 

Abstract - reconsider to add a space between the elevation values and "m" unit. I.e. 1000 m. In the next section (introduction) such spacing is present.

Introduction

78- reconsider changing the phrase "controls soil erosion". Perhaps the verb can be changed into prevents...

82- by using "slow growth habit..." it was meant that it is connected with tree growth tendency or maybe "slow growing habitats"? It is not quite clear. Please rephrase it.

92- "north-western Himalaya", please note that in lines 55-56 it was used the abbrev. of NW Himalayas. Please unify the phrases. See also l. 95, 104 etc.

Materials and Methods

110-111- the phrase: " The soil of the region is silty clay loam..." I would rather write: The soil texture of the region is...

Moreover, does the mentioned soil texture was present in all plots in all elevations?

114 - please use abbrev. of genus Juniperus (as before...) J. macropoda

117 - add a comma after J (abbrev. of Juniperus)

122- E8 (>4260 masl) add a space between m and asl.

125-126- the phrase "However, for regeneration survey, 2 m × 2 m plots were laid out at four corners of the main plot to record the number of seed-lings (<0.5 m) and saplings (0.5-2 m) of J. macropoda." is quite surprising as it does not fits the Figure 1. In this text, you mention about 4 additional 2x2 plots placed in the corners of the "main plot", but in Fif. 1 there are five "inner plots" with the 5-th in the middle. Please explain it clearly.

128- the phrase "The seed bank was also studied in a 1m ×1 m plot." Where this plot was placed? Was it studied in the 4 "inner plots" or in the middle inner plot? This must be clearly stated.

129- add spacing before 1 in the phrase "m ×1 m plot"

138-143- once it was written belowground and with a hyphen below-ground. Please unify the phrase.

Figure 1 - in the legend there is written: "Elevation (m masl)" Is it correct? Moreover, please explain the existence of the fifth "middle inner plot" within the main one (please see the comments above).

Reconsider also changing the title of Figure into i.e. Map of the study area

157- the abbreviation of soil organic carbon should be written as SOC, especially this abbrev. was used in the abstract. Or unify it as OC.

169- I think that between two sentences there is double spacing.

Results

192- please write the (DBH) without dots.

198- correct didn't into did not.

214- add space after followed by...

219 and 233- please add a paragraph when you start the text in the new subsection.

219-255 - please unify the spelling of elevations in the brackets. Once it is written with only the unit m, and the other time with the abbrev. asl

See also the discussion section.

241- OC or SOC and so on.

250-251- the phrase: "Based on the sand, silt, and clay content, the textural class of J. macropoda was classified as sandy loam." This phrase partially answers my previous question (see comment on lines 110-111). But, in the description of the study area (lines 110-111) it was written that the soil of the region consists of silty clay loam. Overall, such a statement should be explained i.e in the description of the study area by stating that the soil texture changes with the elevation. Can you add such information to the introduction? Now the text in lines 110-111 can misleads. Maybe you can add references if there are such.

Figure 2- Average Diameter and Basal Area should be written without a capital letter in the second words i.e. Average diameter etc. Please see and unify with the figure caption.

Figure 3 - as above. Please take a look at Total Biomass.

Figure 4- as above. Please take a look at Total carbon density (in the caption).

Table 2- OC or SOC?, and add a space in the phrase AvailableN

Figure 5-  CaCO3 should be written in lowercase. See line 311 i.e.

Table 3- please check the spacing between the words in the table.

Discussion

357- correct into in situ

388 - SOC or OC - unify the whole manuscript.

440- remove double spacing after i.e.

Conclusions

453 - the phrase north-western was written in two different ways in the manuscript (once with a hyphen and the other times without i.e. Fig. 4, Tab. 2 and so on. Please check it.

Author Response

(The authors gave the same response as above.)

Round 2

Reviewer 1 Report

the Manuscript is suitable to publish
